# Cardiac Insulin Resistance in Heart Failure: The Role of Mitochondrial Dynamics

**DOI:** 10.3390/ijms20143552

**Published:** 2019-07-20

**Authors:** Masao Saotome, Takenori Ikoma, Prottoy Hasan, Yuichiro Maekawa

**Affiliations:** Internal Medicine III, Hamamatsu University School of Medicine, 1-20-1 Handayama, Higashi-ku, Hamamatsu 431-3192, Japan

**Keywords:** cardiac insulin resistance, heart failure, type 2 diabetes mellitus, mitochondrial dynamics

## Abstract

Heart failure (HF) frequently coexists with conditions associated with glucose insufficiency, such as insulin resistance and type 2 diabetes mellitus (T2DM), and patients with T2DM have a significantly high incidence of HF. These two closely related diseases cannot be separated on the basis of their treatment. Some antidiabetic drugs failed to improve cardiac outcomes in T2DM patients, despite lowering glucose levels sufficiently. This may be, at least in part, due to a lack of understanding of cardiac insulin resistance. Basic investigations have revealed the significant contribution of cardiac insulin resistance to the pathogenesis and progression of HF; however, there is no clinical evidence of the definition or treatment of cardiac insulin resistance. Mitochondrial dynamics play an important role in cardiac insulin resistance and HF because they maintain cellular homeostasis through energy production, cell survival, and cell proliferation. The innovation of diagnostic tools and/or treatment targeting mitochondrial dynamics is assumed to improve not only the insulin sensitivity of the myocardium and cardiac metabolism, but also the cardiac contraction function. In this review, we summarized the current knowledge on the correlation between cardiac insulin resistance and progression of HF, and discussed the role of mitochondrial dynamics on the pathogenesis of cardiac insulin resistance and HF. We further discuss the possibility of mitochondria-targeted intervention to improve cardiac metabolism and HF.

## 1. Introduction

Heart failure (HF) is a condition in which the myocardium is unable to meet the whole body’s demand for blood and oxygen and is caused by complex and multiple factors. Because of its high morbidity and mortality, HF is becoming one of the most critical health hazards around the world. An estimated global prevalence of HF may exceed 26 million, and the number of new patients with HF is expanding [1]. To minimize the increasing medical costs among patients with HF, further efforts are required to shorten the hospitalization period and prevent HF recurrence. For that purpose, more effective medicines and/or treatments toward HF are required to innovate both bench and bedside research.

HF frequently coexists with conditions associated with glucose insufficiency, such as insulin resistance or type 2 diabetes mellitus (T2DM), and T2DM patients have a higher incidence of HF than normal subjects. However, the intensive glucose lowering therapy cannot always improve the prognosis of T2DM patients [2], and some antidiabetic drugs failed to improve cardiac outcomes in T2DM patients, despite sufficiently lowering glucose levels [2]. This may be due to a lack of understanding and/or knowledge of cardiac insulin resistance. Although cardiac insulin resistance is considered to develop independently of systemic insulin resistance, the alterations due to systemic insulin resistance, such as overnutrition, oxidative stress, neurohormonal imbalance and increased cytokines significantly contribute to the promotion of cardiac insulin resistance [3]. In this basic research field, there is obvious evidence that cardiac insulin resistance significantly contributes to the pathogenesis and progression of HF [4]. However, currently, neither the definition nor the treatment of cardiac insulin resistance in patients with HF and/or T2DM has been provided.

Mitochondria are organelles that can alter their morphology and location according to mitochondrial dynamics (fusion and fission). Mitochondrial fusion, which is regulated by mitofusin 1 (Mfn1), mitofusin 2 (Mfn2), and optic atrophy-1 (OPA1), supports the intraorganellar exchange of proteins, substrates, and mitochondrial DNA to enhance the stability of the mitochondria. Emerging evidence suggests that an altered mitochondrial morphology (elongated or fragmented) is correlated with the pathophysiology of cardiac diseases and metabolic insufficiency [5]. Accordingly, the development of diagnostic and/or therapeutic tools targeting aberrant mitochondrial dynamics may lead to a preferable result for cardiac insulin resistance, which may contribute to a better prognosis of HF. 

In this review, we summarize the current knowledge regarding the relationship between cardiac insulin resistance and the dysregulation of mitochondrial dynamics in patients with HF and discuss the possibility of a mitochondria-targeted diagnosis and interventions in cardiac insulin resistance. 

## 2. Correlation between HF and Glucose Insufficiency 

### 2.1. Common Coexistence of T2DM in HF Patients

Regardless of the phenotypes (HF with reduced ejection fraction: HFrEF, HF with mid-range ejection fraction: HFmrEF, or HF with preserved ejection fraction: HFpEF), HF is well known to coexist with T2DM [6]. The population-based cohort study (Reykjavik study held during 1967–1997) [7] revealed that the prevalence of HF in T2DM patients was significantly higher (12%, odds ratio was 2.8, 95% CI 2.2–3.6) than those with normal glucose levels (3.2%). Previous major clinical trials reported that the prevalence of HF in T2DM patients is approximately 10% to 30% [6]. Conversely, HF is also well associated with T2DM [6], although the prevalence of T2DM in HF markedly differs by region. The previous major clinical trials on chronic HF indicated that the prevalence of T2DM was approximately 30%, irrespective of the HF phenotypes.

This common coexistence may be responsible for their incidences. In the Kaiser permanent study, a significant higher incidence of T2DM was seen in patients with HF than those without HF (13.6% vs. 9.2%) over five years of observation [8]. In addition, a Danish nationwide cohort study suggested that a significant number of HF patients (8%) developed T2DM over a three-year follow up [9]. The severity of HF is also associated with a risk of T2DM development. Previous clinical trials clearly revealed that the higher the severity of HF, as indicated by the New York Heart Association class, the higher the risk of T2DM development [10,11]. A population-based investigation, which studied 1.9 million T2DM patients (without cardiovascular events), recently demonstrated the increased incidence of HF (14.1%), which is more frequent than vascular events such as stroke and myocardial infarction [12]. Higher HbA1c levels in T2DM patients were associated with a higher incidence of HF and hospitalization than those with lower HbA1c levels [13,14]. 

Furthermore, while HF affects the severity and/or risk of T2DM, the reverse is also true. The comorbidity of patients with HF and T2DM have been reported to worsen HF symptoms than those without T2DM, despite having similar ejection fractions as shown by echocardiograms [15,16]. T2DM also increased the all-cause mortality in patients with HF and readmission with HF [16,17]. In addition, higher HbA1c levels in patients with HF is associated with higher mortality and cardiovascular hospitalization [6]. The CHARM (candesartan in heart failure—assessment of mortality and morbidity) trial revealed that a 1% increase in HbA1c elevated the risk of cardiovascular mortality (hazard ratio 1.1) in patients with T2DM and HF (both HFrEF and HFpEF) [18]. In addition, the GISSI (gruppo Italiano per lo studio della sopravvivenza nell’infarto miocardico; Italian group for the study of the survival of Myocardial Infarction)-HF study indicated that a higher HbA1c was an independent predictor of the composite outcome of mortality and cardiovascular hospitalization [19].

Thus, HF and T2DM coexist frequently and T2DM affects not only new-onset HF, but also morbidity associated with HF.

### 2.2. Antidiabetic Agents and HF Outcome in T2DM Patients

Although insulin resistance and T2DM are strong risk factors for HF, intensive glycemic control using several types of anti-diabetes drugs is known to increase the risk of hospitalization for HF [2]. Therefore, new anti-diabetes drugs must be subjected to evaluation for their cardiovascular safety in clinical trials before approval. 

Treating the impaired insulin sensitivity seems to be effective against both T2DM and HF; however, the outcome differs depending on the type of insulin-sensitizing agent [20,21]. The use of thiazolidinediones (peroxisome proliferator-activated receptor-gamma; PPAR-χ agonists) and both rosiglitazone [21] and pioglitazone [22] led to more HF hospitalizations than the use of a placebo [21,23] according to a randomized clinical trial, because PPAR-χ agonists may enhance renal sodium retention [24], which results in worsening HF. In contrast to PPAR-χ agonists, another insulin-sensitizing agent, metformin, has been recommended as a first-line treatment for patients with T2DM who have cardiovascular risks [25,26]. Although no randomized clinical trials on administering metformin for T2DM patients with HF are currently available, large observational studies revealed that metformin was associated with a lower risk of hospitalization for HF [27]. 

Glucagon-like peptide-1 (GLP-1) receptor agonists enhance insulin secretion and sensitivity. Although GLP-1 receptor agonists expected the cardioprotective effects due to the abundant expression of GLP-1 receptor in human myocardium, clinical trials revealed no improvements in HF hospitalization [6], whereas improvements in cardiac contraction [28] and a significant reduction in cardiovascular death were seen [29]. 

Recent investigations have revealed that sodium–glucose cotransporter 2 inhibitors (SGLT2i) significantly reduced the risk of HF hospitalization in T2DM patients [30,31,32]. The precise mechanisms underlying the beneficial effects of SGLT2i in patients with HF have not been fully elucidated; however, it is considered that multiple extra-myocardial factors, such as osmotic diuresis, blood pressure reduction, sympathetic nerve suppression, and renal protection by SGLT2i are involved [33], rather than direct effects of the failing myocardium, as SGLT2 is not expressed in the myocardium, even under pathophysiological conditions [34]. 

Thus, because not all anti-diabetic agents have reduced the risk of HF, it seems that lowering glucose or insulin sensitization cannot simply reduce the risk of HF in T2DM patients. 

## 3. Cardiac Metabolic Deficiency under HF

### 3.1. Cardiac Energy Metabolism in the Physiological Condition

To maintain a continuous contraction, human hearts are required to produce a high amount of energy phosphate; 3.5–5 kg/day adenosine triphosphate (ATP) is produced [35] from various substrate such as fatty acids (FAs), carbohydrates (glucose and lactate), ketones, and amino acids. In the physiological condition, fatty acids are major substrates that provide approximately 40% to 60% of the total energy, whereas glucose contributes less than 25% to myocardial energy production [36]. However, hearts can rapidly alter their substrate use depending on the hormonal status, workload of the heart, and circulating substrate condition on the basis of metabolic flexibility [35].

Glucose enters cardiac myocytes through glucose transporter 1 (GLUT1) and 4 (GLUT4); GLUT1 is insulin-independent and located on the cellular membrane, and GLUT4 is translocated to the cellular membrane in an insulin-dependent manner. Intracellular glucose is initially phosphorylated by hexokinase to glucose-6-phosphate (G6P), which is a substrate of glycolysis, to produce pyruvate, NADH, and 2ATP. Pyruvate that derives from glycolysis is either converted to lactate via lactate dehydrogenase (LDH) or transferred to the mitochondria (via the mitochondria pyruvate carrier: MPC) and oxidized by the Krebs cycle (Figure 1) [36].

Human hearts prefer to utilize long-chain fatty acids that bind to albumin and/or are released from triacylglycerol contained in chylomicrons or very-low-density lipoproteins (VLDL) [35]. Circulating free fatty acids (FAs) enter cardiac myocytes through transport proteins (fatty acid translocase; FAT-CD36; plasma membrane fatty acid-binding protein (FABPpm), and fatty acid transport proteins 1 and 6 (FATP1 and FATP6)) or by passive diffusion [36,37]. Cardiac FA uptake via FAT-CD36 is stimulated by insulin or AMP-activated kinase (AMPK) [37]. After entering into the cytosol, FAs are esterified into long-chain fatty acyl CoA [38] and directly shuttled to the mitochondria to start β-oxidation. The long-chain fatty acyl CoA is converted to long-chain acylcarnitine by carnitine palmitoyltransferase isomers-I (CPT-I) located in the outer mitochondrial membrane (OMM) and then converted back to long-chain fatty acyl CoA by palmitoyltransferase isomers II (CPT-II) located in the inner mitochondrial membrane (IMM) [35,36]. 

### 3.2. Cardiac Insulin Resistance and Impaired Cardiac Metabolic Flexibility under HF

HF is a syndrome in which hearts fail to respond to systemic demands for many etiological reasons, and cardiac energy production is generally impaired regardless of the type and/or stage of HF. However, the exact energy metabolic profile of HF remains controversial, because there is still a discrepancy regarding the substrate preference where the oxidation rate of substrates (fatty acid, glucose, and ketone body) varies by model and/or stage of heart failure [36]. 

It is generally believed that a failing heart, in which oxygen is deficient, alters the main source of energy where the heart is less dependent on FA oxidation and switches to glycolysis as its main source of energy. Indeed, under pathophysiological conditions such as HF, glycolysis and glucose oxidation support the myocardial energy demand, as a small amount of ATP is obtained by substrate-level phosphorylation via glycolysis and may be important for the maintenance of intracellular ionic homeostasis even in a reduced O_2_ environment. In addition, impaired mitochondrial TCA (tricarboxylic acid cycle) activity and decreased enzyme transcription involved in FA oxidation have been reported under various types of HF, such as a pressure-overload and a rapid pacing-induced HF model [39]. 

In contrast to FA oxidation in HF, glucose metabolism is varied by its etiology and stage of HF [40]. In fact, cardiac glucose uptake varies with experimental models; e.g., glucose uptake was decreased in the HF model by aortic constriction, whereas increased in the HF model by Dahl salt-sensitive rat [40]. Emerging bodies of evidence have revealed a strong relationship between HF and insulin resistance [41]. Because both glucose and fatty acid metabolism in the heart are tightly controlled by insulin signaling, insulin resistance plays a critical role in the pathogenesis of HF. Indeed, an impairment in insulin signaling and the development of insulin resistance, which frequently proceed cardiac dysfunction in heart failure, are major determinant factors of HF progression [42]. Insulin resistance and/or T2DM increases the delivery of FAs to the myocardium. Previous investigations showed increased myocardium lipid accumulation in T2DM patients by cardiac biopsy [43] and cardiac magnetic resonance imaging (MRI) [44], and the accumulation of lipid metabolites such as triacylglycerol, ceramide, and diacylglycerol—which can induce further metabolic deficiency and/or apoptosis—is toxic to cardiac myocytes [36,37,38]. In addition, it is well known that cardiac metabolic flexibility is impaired [45,46] in some cardiac diseases, which consequently reduces myocardial ATP production [47] and cardiac function.

### 3.3. Molecular Mechanism of Impaired Insulin Signaling under HF

In HF, the compensatory hyper-adrenergic state can increase the circulating plasma FAs, which are derived from adipose tissue (Figure 2). The excess circulating FAs inhibit glucose uptake by impaired insulin signaling in skeletal muscle (and cardiomyocytes) and, thereby, increase blood glucose by insulin resistance. The increased circulating FAs also lead to excessive ROS generation and oxygen waste through activated beta-oxidation in the mitochondria, which leads to further adverse effects for preexisting impaired cardiac contractility. Thus, these metabolic alterations (such as increased circulating FAs, hyperglycemia, and hyperinsulinemia) induced by the compensatory hyperadrenergic state in HF further deteriorate cardiac function and should be corrected via metabolic intervention.

## 4. The Role of Mitochondrial Dynamics in Cardiac Insulin Resistance and HF

### 4.1. Mitochondrial Energy Production in a Normal Heart

Under physiological conditions, the mitochondria produces most of the ATP (>95%) that supports the intracellular energy demands of cardiac muscle through oxidative phosphorylation (OXPHOS) in the inner mitochondrial membrane (IMM) in a group of proteins known as the electron transport chain (ETC) [35]. In OXPHOS, ATP synthesis (F_1_-F_0_ ATPase) essentially depends on a potent electrical and proton gradient across the mitochondrial membrane (mitochondrial membrane potential; ΔΨ_m_ = −180 ~ −200 mV). 

Because mitochondrial Ca^2+^ plays an important role in ATP synthesis [48], mitochondrial Ca^2+^ regulation is important for various cellular physiological processes. Classically, mitochondria accumulate Ca^2+^ through the mitochondrial Ca^2+^ uniporter by primarily using the electrical gradient (ΔΨ_m_) [49]. To respond to the beat-to-beat base energy demand, mitochondrial Ca^2+^ is mobilized from the sarcoplasmic reticulum (SR) [50] through a physical coupling between the SR to the mitochondria [51]. Mitochondrial Ca^2+^ efflux in the heart is mainly conducted by the mitochondrial Na^+^/Ca^2+^ exchanger (NCLX) and by the mitochondrial permeability transition pore (mPTP) [49,52]. 

Mitochondria are known as the major organelles that produce reactive oxygen species (ROS) through mitochondrial ETC activity. However, myocardial ROS remains at a low level under the physiological conditions, because of effective mitochondrial antioxidant systems, such as manganese superoxide dismutase (MnSOD), catalase, and glutathione peroxidase (GPx) [53].

### 4.2. Mitochondrial Dysfunction Induces Cardiac Insulin Resistance in Patients with HF

Emerging bodies of evidence have revealed that mitochondrial dysfunction is a key player in pathogenesis in insulin resistance and heart failure [54,55]. In patients with HF, mitochondrial dysfunction induces excess ROS production, reduced mitochondrial membrane potential, impaired ETC complex activity, and abnormal mitochondrial biogenesis, which thereby results in decreased ATP synthesis [47,56] and cardiac remodeling [55] (Figure 2). Mitochondrial ROS elevation is promoted through a self-amplifying loop called ROS-induced ROS release (RIRR); that is, the initial mitochondrial ROS elevation can induce the consequent mitochondrial ROS burst. As RIRR is associated with the mitochondrial permeability transition pore (mPTP) opening and consequent ΔΨ_m_ dissipation [52,57], the myocardial ROS burst during the reperfusion phase might result from RIRR. 

In patients with insulin resistance and T2DM, the reduced mitochondrial OXPHOS—due to mitochondrial dysfunction—promotes excess intracellular lipid metabolite accumulation (such as fatty acyl CoAs and diacylglycerol) in the skeletal muscle [58]. The intracellular lipid metabolite accumulation activates serine/threonine kinases, such as protein kinase C (PKC-θ rodents, PKC-β and -δ humans) [59,60], which in turn produce phosphorylate serine residues on IRS-1, inhibit insulin-induced PI 3-kinase activity, and lead to reduced AKT (protein kinase B) 2 activity after insulin stimulation. Impaired AKT2 activity suppresses insulin-mediated GLUT4 translocation to the plasma membrane, which consequently compromises insulin-mediated glucose uptake.

Thus, mitochondrial dysfunction plays a key role in the pathogenesis of HF and insulin resistance, and targeted treatment to the mitochondria has been considered to improve HF and insulin resistance.

### 4.3. Dysregulation of Mitochondrial Dynamics in HF

To respond to the change of cellular demand, mitochondria can alter their morphology through fusion and fission events (so-called mitochondrial dynamics), and the dysregulation of mitochondrial dynamics is known to be correlated with the pathogenesis of cardiac diseases [5]. Mitochondrial fusion, which is regulated by mitofusin 1 (Mfn1), mitofusin 2 (Mfn2), and optic atrophy-1 (OPA1), supports the inter-organellar exchange of proteins, substrates, and mitochondrial DNA to enhance the stability of mitochondria [61]. Mitochondrial fission, which is regulated by mitochondrial fission proteins, such as dynamin-related protein-1 (DRP1), mitochondrial fission factor (MFF) [62], and fission 1 homologue protein (Fis1) [63], increases the number and capacity of mitochondria during cell division. In addition, mitochondrial fission also facilitates the control of mitochondrial quality by mitophagy, which removes damaged mitochondria through lysosomal autopsy.

Evidence suggests that the alteration in mitochondrial morphology correlates with the pathophysiology of insulin resistance and heart failure [64,65]. In the failing myocardium, mitochondrial fission was apparent, and the association of both mitochondrial dynamic proteins (decrease in fusion proteins and/or increase in fission proteins) with a failing heart was reported [66,67]. In addition, it has been reported that small mitochondria were frequently seen in the skeletal muscle in obese and T2DM subjects [68,69] as well. On the other hand, chronic oxidative stress is known to play the most crucial role for the pathogenesis of insulin resistance [70]. We have reported that DRP1 and ROS have a mutual enhancing relationship, in which DRP1 enhances ROS generation, and vice versa [71]. Although the underlining mechanism remains elusive, the increase in DRP1 and mitochondrial fragmentation are interlinked with ROS. This reciprocal augmentation of DRP1 and ROS could be a key player which induces mitochondrial dysfunction and the ultimate fate of myocardial insulin resistance.

## 5. Cardiac Insulin Resistance in Patients with HF

### 5.1. Diagnosis of Cardiac Insulin Resistance in Patients with HF and DM

Despite this significant contribution of cardiac insulin resistance in the basic investigations of HF, there is no current available definition and/or diagnosis criteria of cardiac insulin resistance. Insulin resistance is clinically defined as a condition of impaired insulin sensitivity to glucose uptake and/or utilization. Systemic (whole body) insulin resistance has been defined by the “homeostasis model assessment for insulin resistance (HOMA-IR: normal; <1.6, insulin resistance; >2.5)”, which is calculated as follows: fasting plasma glucose (mmol/L) × fasting serum insulin (IU/L)/405 [72]. However, currently, neither relevant clinical diagnostic criteria nor clear reference values of cardiac insulin resistance have been established.

The ^18^F-fluorodeoxyglucose positron emission tomography (FDG-PET) is known to be a clinically extant method to assess myocardial glucose uptake. The insufficiency of cardiac glucose uptake has been investigated using the hyper-insulinemic/euglycemic clamp method, which was developed by the American Society of Nuclear Cardiology and can indicate maximal myocardial glucose uptake [73]. However, so far, FDG-PET remains in the relative comparison (with normal subjects and/or among disease groups) [74] and fails to provide any clear cutoff values to diagnose cardiac insulin resistance. The assessment of cardiac insulin resistance may require at least two independent points of myocardial FDG uptake, i.e., fasting (6–12 h fasting to minimize the heterogeneity of FDG accumulation) and maximal glucose uptake (by glucose and, if feasible, insulin loading). Because the bioactive half-life of ^18^F-fluorodeoxyglucose (^18^F-FDG) is approximately 110 min [75], multiple consecutive scans are technically impossible and single scans are applied even in the evaluation of coronary artery disease (myocardial ischemia). Thus, the development of a glucose tracer that enables temporal assessment (fasting and glucose with insulin loading) and spatial dynamic FDG-PET is desired to assess cardiac insulin resistance. 

Cardiac biopsies and the investigation of insulin signaling molecules, such as insulin receptor substrate 1 (IRS1), IRS1-associated PI3K (IRS1-PI3K), and GLUT4, may be alternative and direct diagnostic methods to evaluate myocardial insulin resistance [76]. The hyper-phosphorylation of IRS1 and/or IRS1-PI3K reduced the expression of GLUT4 in the myocardium, which may reflect cardiac insulin resistance. However, this also remains in relative comparison with normal subjects. In addition, Cook et al. reported that glucose utility was reduced both in patients with T2DM and LV dysfunction, whereas IRS1-PI3K was unexpectedly elevated, and concluded that behaviors of insulin signaling molecules in cardiac insulin resistance are different from those of whole body insulin resistance [76]. Because degraded mitochondrial morphology frequently accompanies myocardial metabolic disorders in patients with HF [64,65], it seems possible that the assessment of mitochondrial morphology and/or size in electron microscopy may at least partially support a diagnosis of cardiac insulin resistance and metabolic disorder. Further investigations are required to reveal the relationship between aberrant mitochondrial morphology and cardiac insulin resistance in a clinical study.

### 5.2. Innovation of Treatments to Improve Cardiac Insulin Resistance

Although an improvement in cardiac insulin resistance is anticipated to afford preferable effects on HF, clinically established treatment toward cardiac insulin resistance is not currently available. Evidence on the validated anti-diabetic agents may give us some tips on developing new medication for cardiac insulin resistance. As mentioned above, there is a discrepancy regarding the cardiac outcome of insulin-sensitizing agents (see Section 2.2). Because thiazolidinediones, which are considered a contraindication for patients with HF [21,23], are PPAR-χ agonists, they can activate the uptake and oxidation of FAs (simultaneously with glucose) and further enforce FA uptake and oxidation in a failing heart. The enhancement of FA metabolism under HF, which may exceed the capacity of impaired mitochondrial respiration, can lead to the accumulation of lipid metabolites [37], accelerate dysregulated mitochondrial dynamics [71], which enhance cardiac insulin resistance, and myocardium energy shortage, and thereby deteriorate HF [65,77]. In contrast, metformin—which has been recommended as a first-line treatment to T2DM patients who have cardiovascular risks [25,26]—enhances energy expenditure, suppresses hepatic gluconeogenesis, and induces insulin sensitization through AMP-activated protein kinase (AMPK) activation. Activated AMPK can also suppress the mitochondrial fission protein of DRP1, which can resist lipotoxicity-induced mitochondrial dysfunction [77], and ameliorate cardiac insulin resistance. 

Mdivi-1 is a selective DRP1 (mitochondrial fission protein) inhibitor. A number of investigations have reported the preferable effects of mdiivi1 on ischemia/reperfusion injury [5] and cardiac hypertrophy in animal models [78]. Little is known about the effect of mdivi1 on T2DM and HF. Furthermore, clinically relevant evidence has been less reported. Recently, Ishikawa et al. formulated poly nanoparticles containing mdivi-1, which could be delivered to the cytosol and mitochondria, and exhibited better cardiac protection against ischemia reperfusion injury than mdivi1 alone through the inhibition of mitochondrial outer membrane permeabilization (MOMP) [79]. On the other hand, there are some reports that suspect a selective inhibitory effect of mdivi-1 on Drp1. Bordt et al. showed that mdivi-1 promotes mitochondrial complex I-dependent O2 consumption and produces reverse electron transfer-mediated ROS with an independent manner of mitochondrial elongation, and concluded that mdivi-1 does not selectively inhibit Drp1 but contributes to mitochondrial morphological alteration and cellular protection by reversible Complex I inhibition and mitochondrial ROS suppression [80]. Elamipretide is a mitochondria-targeting agent, which selectively acts on mitochondrial ECT to improve the efficacy of electron transport. Although there are fewer human clinical trials, a number of animal investigations have revealed promising results on HF [81]. Thus, currently, there is no clinically available treatment targeting mitochondria or mitochondrial dynamics. Further investigations are required to reveal the effects of mitochondrial targeting agents on cardiac insulin resistance and HF.

## 6. Summary

HF frequently coexists with glucose insufficiency such as T2DM and insulin resistance. However, some antidiabetic drugs failed to improve cardiac outcomes in T2DM patients, although glucose levels are sufficiently lowered to cause an effect. This may, at least in part, suggest a lack of understanding of cardiac insulin resistance, as despite the significant contribution of cardiac insulin resistance to the pathogenesis and progression of HF in basic research, there is no information on the definition or treatment of cardiac insulin resistance from a clinical study. 

Because mitochondrial dynamics play an important role in the pathogenesis of HF and insulin resistance, the development of a mitochondria-targeted diagnosis and/or treatment may afford preferable outcomes for patients with HF. Further investigations are encouraged to reveal the relationship between mitochondrial dynamics and cardiac insulin resistance in HF.

## Figures and Tables

**Figure 1 ijms-20-03552-f001:**
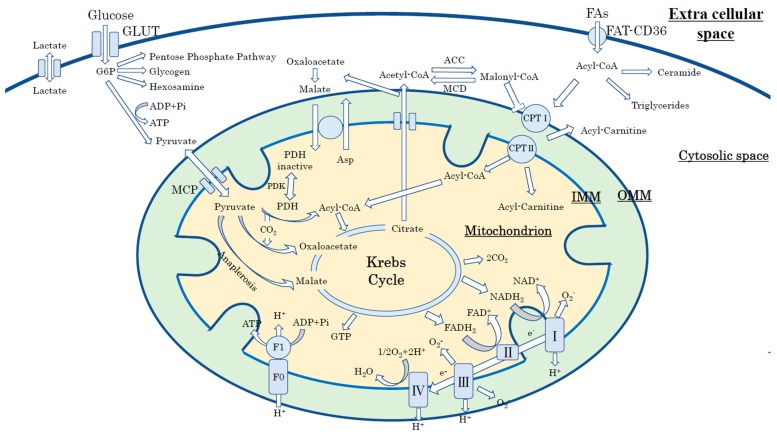
Cardiac energy metabolism in the physiological condition. FAs, fatty acids; GLUT, glucose transporter; FAT-CD36, fatty acid translocase; G6P, glucose-6-phosphate; MPC, mitochondria; OMM, outer mitochondrial membrane; IMM, inner mitochondrial membrane; CPT-I/II, carnitine palmitoyltransferase isomers-I/II.

**Figure 2 ijms-20-03552-f002:**
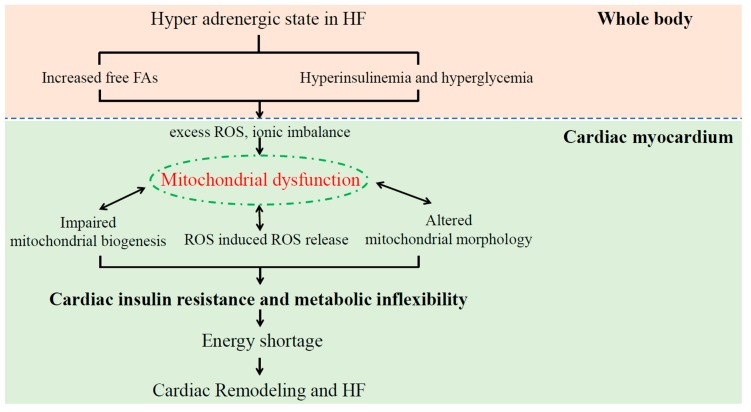
The role of mitochondrial dysfunction in the pathogenesis of cardiac insulin resistance and HF.

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
