# Peer review of "Cardiac Insulin Resistance in Heart Failure: The Role of Mitochondrial Dynamics"

_ijms, 2019, doi:10.3390/ijms20143552_

Round 1
Reviewer 1 Report
In this review, Masao Saotome et al. are introducing the mitochondrial dynamics as a potential actors into the heart failure.
The paper is divided in 6 parts. The three first points describe the current knowledges about heart failure and the cardiac metabolic flexibility impairement mainly correlating to the insulin resistance. The other points are focused on the mitochondrial dynamics via the mitofusins 1/2 and DRP1 regulations and the morphology changes.
This review is clear, nicely written and well-documented. The authors provided 2 pictures helping the review understanding.
This review can be published in that state but it will be useful to include a heart failure description in the introduction.
minor reviews
It will help to the understanding if the authors can include a small heart failure description in the introduction.
Author Response
Dear the reviewer ,
We appreciate your constructive comments.
We have carefully revised the manuscript to accommodate the comments of reviewers. In the revised manuscript, we corrected text with using “Track-change”. In addition, we have received an English check before submission. We hope the revised version of manuscript will be acceptable for the publication in IJMS.
Please find the attached word file.

Reviewer 2 Report
In this review the Authors sought to summarize the current knowledge regarding the relationship between cardiac insulin resistance and dysregulation of mitochondrial dynamics in patients with HF and to discuss the possibility of a mitochondria-targeted diagnosis and interventions in cardiac insulin resistance.
The topic of the review is interesting, even if it lacks of some important data (i.e. the role of liraglutide in improving cardiac outcomes in diabetic patients with HF; see Arturi F., et al. Endocrine 2017).
I suggest to shorten the Introduction section, that is quite redundant in the present form.
Author Response
Dear reviewer ,
We appreciate your constructive comments.
We have carefully revised the manuscript to accommodate the comments of reviewers. In the revised manuscript, we corrected text with using “Track-change”. In addition, we have received an English check before submission. We hope the revised version of manuscript will be acceptable for the publication in IJMS.
Please find the attached word file.

Reviewer 3 Report
Overall Comments:
I found this to be a very interesting and thoughtful review on the connection between heart failure and insulin resistance. I think it requires help from an English language editor, although it is overall quite readable. Also it would be helpful if the authors gave considerably more information about the nature of the conclusions that are discussed and cited (human vs animal study, type of study/model system). Finally, I think the authors need to present a more nuanced view about what is not known and also to discuss critique and alternate interpretations of key hypothesis that are discussed. Detailed critique is below:
Abstract
1) No comments
Section 1
1) Can the authors elaborate a little more on the “altered mitochondrial morphology” that is mentioned?
Section 2.1
1) Is there evidence of causality in the connection between HF and T2DM? Or only correlation?
Section 2.2
1) Is there any evidence of correlation between blood sugar levels or Hemoglobin A1C levels and heart failure onset or progression?
2) For medications that treat T2DM but cause worsened HF, is there evidence that this occurs via some known pharmacology that is unrelated to the effects on glycemic control?
Section 3.2
1) In the discussion of increased FA uptake by myocardial cells in the setting of T2DM to produce toxic lipid species, can the authors elaborate more fully on what experimental models generated this conclusion? Also, can you discuss further the toxic lipid metabolites?
Section 3.3
1) What are the alternative explanations to and/or critique of the “metabolic vicious cycle”? Can the authors present a more balanced view?
Section 4.3
1) One human tissue observational study is described, but can the authors give a better sense of how the data for mitochondrial fragmentation were derived? Are these primarily animal experiment, and if so, what model system(s) were used?
Section 5.2
1) What stage in development is Mdivi-1 in currently? Can the authors give more information?
Author Response

(The authors gave the same response as above.)
